# Research Advances on the Bioactivity of 1,2,3-Triazolium Salts

**DOI:** 10.3390/ijms241310694

**Published:** 2023-06-27

**Authors:** Jia Song, Jie Lv, Jiamiao Jin, Zhichao Jin, Tingting Li, Jian Wu

**Affiliations:** National Key Laboratory of Green Pesticide, Key Laboratory of Green Pesticide and Agricultural Bioengineering, Ministry of Education, Center for Research and Development of Fine Chemicals, Guizhou University, Guiyang 550025, China; jsong21@126.com (J.S.); qwlvjie1130@163.com (J.L.); jjm0058@163.com (J.J.); zcjin@gzu.edu.cn (Z.J.)

**Keywords:** 1,2,3-triazolium salts, synthesis, antibacterial, antifungal, anticancer, antileishmanial, SAR

## Abstract

1,2,3-Triazolium salts have demonstrated significant potential in the fields of medicine and agriculture, exhibiting exceptional antibacterial, antifungal, anticancer, and antileishmanial properties. Moreover, these salts can be utilized as additives or components to produce nano- and fiber-based materials with antibacterial properties. In this review, we summarize several synthetic strategies to obtain 1,2,3-triazolium salts and the structures of 1,2,3-triazolium derivatives with biological activities in the domains of pharmaceuticals, pesticides, and functional materials. Additionally, the structure–activity relationship (SAR) of 1,2,3-triazolium salts with different biological activities has been analyzed. Finally, this review presents the potential applications and prospects of 1,2,3-triazolium salts in the fields of agriculture, medicine, and industrial synthesis.

## 1. Introduction

1,2,3-Triazoles are widely used in the fields of pesticides and medicines due to their broad-spectrum bioactive properties [1]. However, the increasing problem of drug resistance associated with the frequent use of triazole drugs needs to be addressed through the development of novel pharmacophores [2]. The 1,2,3-triazolium unit is a significant bioactive group, which has attracted much attention in recent years [3,4]. Studies have shown that the triazolium transformated from triazole have good bioactivities [5,6,7,8,9,10,11,12]. Molecules containing 1,2,3-triazolium units have been proven to exhibit a variety of biological activities, including antibacterial [13,14,15,16,17,18,19,20,21,22,23,24,25,26,27,28,29,30,31,32,33,34], antifungal [5,6,7,8,9,35], anticancer [10,11,36,37,38,39,40,41,42], and antileishmanial [12,43,44,45,46] properties.

1,2,3-Triazolium salts possess unique chemical properties, such as high stability, tunable reactivity, and distinct electronic characteristics [4,47,48]. They are valuable building blocks in organic synthesis, and also have been employed in the development of novel catalysts, antimicrobial agents, fluorescent dyes, and bioactive molecules [47,48,49,50]. For example, functionalized biopolymers containing 1,2,3-triazolium units have shown excellent potential to produce fibers with antibacterial properties, dental materials, and medical antibacterial devices [32,33]. In general, 1,2,3-triazolium salts could be obtained through the copper-catalyzed 1,3-dipolar azide-alkyne cycloaddition (CuAAC) reaction, *N*-alkylation, and salt metathesis processes [47,48]. 1,2,3-Triazolium salts have three active sites, including N-1, N-3 and C-4 positions, which can be linked with different functional groups in a regioselective manner to regulate their biological activity and cytotoxicity [4].

Therefore, it is necessary to summarize the structure of triazolium salts with different biological activities and the structure–bioactivity relationship (SAR) in the past 25 years. This review is divided into five main sections: (1) synthesis of 1,2,3-triazolium salts; (2) antibacterial activity of 1,2,3-triazolium salts; (3) antifungal activity of 1,2,3-triazolium salts; (4) anticancer activity of 1,2,3-triazolium salts; (5) antileishmanial activity of 1,2,3-triazolium salts. We hope that this review can inspire the further design and synthesis of 1,2,3-triazolium salt structures for various bioactive applications.

## 2. Synthesis of 1,2,3-Triazolium Salts

In the early 21st century, a breakthrough in triazole chemistry occurred when the Sharpless group first proposed a new concept known as “click” chemistry [51]. “Click” chemistry is a powerful synthetic strategy that aims to efficiently and selectively generate chemical bonds under mild reaction conditions [47,48,49,50,51,52]. One of the most versatile click reactions is the synthesis of 1,2,3-triazolium salts [52,53]. The click chemistry for the synthesis of 1,2,3-triazolium salts generally includes three steps using azide and alkyne as the starting materials: cycloaddition reaction of 1,2,3-triazole, *N*-alkylation, and salt metathesis [47,48,50].

### 2.1. Non-Selective Huisgen Reaction

The utilization and synthesis of 1,2,3-triazoles have experienced significant growth in recent decades [52]. The existence of triazoles has been recognized since the 1960s when Huisgen discovered the thermal cycloaddition reaction involving azides and alkynes (Figure 1) [54]. Under high temperatures, the Huisgen 1,3-dipolar cycloaddition reaction between azides and terminal or internal alkynes leads to the formation of triazoles [54]. However, this method is limited due to its requirement for prolonged heating and its purification from the mixture of 1,4- and 1,5-regioisomers [52,55].

### 2.2. Selective Huisgen Reaction Mediated by Metal Catalysis

In 1974, Huisgen and coworkers found that Cu(I) cations could catalyze the cyclization reaction between an azide and an alkyne, leading to the formation of 1,2,3-triazoles [4]. They demonstrated that under catalytic conditions, the CuAAC reaction could selectively generate the 1,4-disubstituted isomer as the main product [4]. The key step in the Huisgen synthesis is the formation of a copper(I) acetylide intermediate from the alkyne and the copper catalyst [55,56,57]. This intermediate then reacts with the azide to generate a highly stable 1,2,3-triazole product. The reaction exhibits high regioselectivities under mild reaction conditions, giving the 1,4-disubstituted triazole product as the major isomer (Figure 1) [55,56,57]. The selective Huisgen synthesis offers several advantages over traditional methods for triazole synthesis [55]. It proceeds rapidly, often achieving complete conversion within minutes to hours. It also tolerates a wide range of functional groups, enabling the incorporation of diverse molecular entities into the triazole scaffold. Additionally, the reaction can be performed in water or bio-orthogonal conditions, making it suitable for applications in biological systems [55].

In 2008, Fokin’s group chose Ruthenium (II) as a catalyst to promote the reaction between a series of organic azides and terminal alkynes containing various functional groups. 1,5-disubstitued 1,2,3-triazole products were selectively generated at a high temperature (Figure 1) [58]. Ru(II) catalysts possess advantages over copper catalysts due to their easy synthesis and air stability, which are suitable for ambient temperature cycloaddition reactions involving internal alkynes, aryl azides, and other thermally labile reactants [58]. Nonetheless, both catalysts exhibit excellent activities as well as chemo- and regio-selectivities. In addition to the CuAAC reaction, the new RuAAC process provides easier access to all 1H-1,2,3-triazole regioisomers [57,58].

### 2.3. N-Alkylation and Salt Metathesis

The highly efficient Cu(I)-catalyzed azide-alkyne “click” cycloaddition (CuAAC) can provide facile access to 1,4-disubstituted 1,2,3-triazoles and make them widely-used scaffolds in the field of synthetic chemistry [55,56]. Structurally complex 1,2,3-triazolium salts can be readily prepared from the corresponding 1,2,3-triazoles using alkyl halides, tosylate, trifluoromethane sulfonate, or tri-alkyloxoniums (Figure 2) [49,50]. The alkylation process typically involves the use of a base to deprotonate the 1,2,3-triazole, followed by a reaction with the alkylating agent. This transformation generally tolerates a wide variety of functional groups at the N-1 and C-4 positions of the triazole ring and often occurs with generally N-3 regioselectivity [49,50].

The resulting alkylated product can be further converted into the various 1,2,3-triazolium salts through protonation or appropriate transformations [4,50]. For example, the anionic counterion X^−^ of the triazolium salt can be easily exchanged by simply washing with excess inorganic salt or using an exchange resin [17] (Figure 3).

## 3. Antibacterial Activities of 1,2,3-Triazolium Salts

Quaternary ammonium compounds (QACs) are a kind of cationic biocide with broad-spectrum antibacterial activity which are used in various fields from household cleaning and agriculture to medicine and industry [59]. QACs possess unique amphiphilic characteristics that allow them to disrupt the phospholipid bilayer of the cell membrane [59]. Consequently, they are often used as bactericides, such as benzalkonium chloride [60]. 

### 3.1. Antibacterial 1,3,4-Trisubstituted-1,2,3-triazolium Salts

Analogues of the functional 1,3,4-trisubstituted-1,2,3-triazolium bromide salts have demonstrated excellent antibiotic properties. In 2018, Fletcher et al. synthesized a series of 1,3,4-trisubstituted-1,2,3-triazolium bromide salts through a two-step process involving azide-alkyne cycloaddition and benzyl substitution [13]. Several of these compounds exhibited good antibacterial activities against gram-positive (G+) strains and gram-negative (G-) strains. Compound **1** demonstrated the best antibacterial activity, with minimum inhibitory concentration (MIC) values of less than 8 µM. In addition, it displayed a good inhibitory effect against yeast, with a MIC value of 8 µM against *Saccharomyces cerevisiae* (*S. cerevisiae*) (Figure 1). 

Four years later, they synthesized more 1,3,4-trisubstituted-1,2,3-triazolium bromide salts via click reactions [15]. Compounds with 2-fluorenyl, 1-naphthyl, 2-naphthyl, 2-anthracenyl, or 1-pyrenyl at the N-1 position of the triazolium salt exhibited emission properties and good antibacterial activities. Compound **2** showed similar inhibitory activities against G+ strains as compound **1**, with MIC values of 0.4–0.8 μM (Figure 1).

In 2021, Sol and coworkers prepared a variety of 1,2,3-triazolium-containing phenolic ketone derivatives with moderate to good water solubility [14]. Among them, compound **3** exhibited good resistance to bacteria in the absence of light, and its antibacterial activity could be further increased under light irradiation through the light-induced activation of the highly conjugated R group. The MIC values of compound **3** against *Staphylococcus aureus* (*S. aureus*) CIP76.25 and *Staphylococcus epidermidis* (*S. epidermidis*) CIP109.562 under light conditions were 0.39 µM and 0.78 µM, respectively (Figure 1).

### 3.2. Antibacterial Cationic Anthraquinone Analogs

Anthraquinone is the structural core of anthracycline and exhibits a wide range of biological activities [17]. It has played an important role in the discovery of new biological and pharmaceutical therapeutic drugs [18]. The analogs of anthraquinone fused with 1,2,3-triazolium units are called cationic anthraquinone analogs (CAAs), which have been reported to possess good antibacterial activities [17,18,19]. These compounds are composed of several popular antibacterial scaffolds, including anthraquinone, triazolium, and QAC cations.

Since 2011, the Chang group has systematically studied the biological activities of anthraquinone derivatives derived from 1-alkyl-1*H*- or 2-alkyl-2*H*-naphtho [2,3-d]triazole-4,9-diones to search for new bactericidal molecules against drug-resistant bacteria [16,18]. They found that anthraquinone derivatives with cationic triazolium units possessed good antibacterial activities. For example, compound **4** showed good activity against *S. aureus*, with MIC values of 0.032 to 0.064 μg/mL (Figure 2).

In the same year, to clarify the effects of alkyl chain length on antibacterial activities, they synthesized linear alkyl chain anthraquinone triazolium salts with different sizes at the N-3 position and tested their activities against drug-resistant bacteria [18]. The results showed that compounds **5a** and **5b** had broad activities against methicillin-resistant *S. aureus* (MRSA) and vancomycin-resistant *Enterococcus faecalis* (VRE), and the MIC values were both less than 8 μg/mL (Figure 2).

Later, they used the same method to obtain a series of anthraquinone triazolium salts containing different alkyl chains connecting to the N-1 and N-3 positions. They found that compound **6** (Figure 2) had the best inhibitory activities against *S. aureus* and *Escherichia coli* (*E. coli*), with MIC values of 0.25 to 2 μg/mL [17].

In 2017, they developed a series of dimeric cationic anthraquinone analogs with antibacterial activities [19]. The MIC values of compound **7** against G+ bacteria and G- bacteria were 1 to 16 μg/mL. In the following year, anthraquinone triazolium compounds with different anions were also tested for antibacterial activities [20]. The MIC values of compound **8** with ^¯^OTf anions against *S. aureus* and *E. coli* were 0.125 to 1 μg/mL. In addition to the alkyl chain, the anthraquinone triazolium compounds connecting the aryl group at the N-3 position also showed good inhibitory activities against MRSA. The MIC values of Compounds **9a** and **9b** against MRSA were 1 to 2 μg/mL (Figure 2) [21].

### 3.3. Antibacterial 1,2,3-Triazolium-Based Peptoid Oligomers

Amphiphilic cationic peptides with *N*-substituted glycine showed great prospects as antimicrobial peptide mimetics. In 2018, Faure and coworkers obtained a series of 1,2,3-triazolium cationic amphiphilic peptide oligomers on a carrier [23]. Short hexamer **10** had a great inhibitory effect against G+ bacteria with MIC values of 6.3 μM and 3.1 μM against *E. faecalis* and *S. aureus*, respectively (Figure 3). Furthermore, this hexamer was found to be non-hemolytic and non-cytotoxic to Hela cells.

In 2021, they reported that the hexamer **11**, which contains a triazolium cationic side chain, exhibited good inhibition activities against G+ strains, with a MIC value of 3.1 μM against *S. aureus* CIP 6525 [24]. This hexamer was the first peptide reported to have a preventive effect on biofilm formation in *P. aeruginosa* and *E. faecalis* at sub-MIC concentrations. The hemolytic assay demonstrated that Compound **11** had good selectivity and was neither hemolytic nor cytotoxic to Hela cells, even at high concentrations (Figure 3).

### 3.4. Antibacterial Condensed-Heterocyclic 1,2,3-Triazolium Salts

In 1992, Yoshimura et al. synthesized a series of antibacterial compounds by transforming cephalosporins into heterocycles fused with triazolium [25]. Compound **12**, which contains 1,2,3-triazolium cationic groups, showed effective antibacterial activities against *S. aureus* 308A-1, *E. coli* NIHJ JC-2, *E. cloacae* IFO 12937, *S. marcescens* IFO 12648, *Proteus vugaris* (*P. vugaris*) IFO 3988, and *P. aeruginosa* IFO 3455, with MIC values of less than 3.13 µg/mL (Figure 4).

Recently, Wilson et al. synthesized a series of *N,N*’-disubstituted triazolium salts, and tested their antibacterial activity against several bacteria, including *Enterococcus faecium (E. faecium)*, *S. aureus*, *Klebsiella pneumoniae (K. pneumoniae)*, *Acinetobacter baumannii (A. baumannii)*, *Pseudomonas aeruginosa (P. aeruginosa)* and *E. faecium*, which are common pathogens of hospital-acquired infections [26]. Those compounds exhibited potent antibacterial properties against all bacterial strains. Specifically, compound **13** exhibited a MIC value of ≤0.5 µg/mL (Figure 4).

### 3.5. Antibacterial 1,2,3-Triazolium-Based Polymers

Polymers containing QACs are extensively studied as antibacterial and disinfectant agents [27,28]. The large-scale synthesis of the polymers with QACs can be cost-effective compared with antimicrobial peptides and antibiotics [27]. The antibacterial activity and blood toxicities of copolymers are significantly influenced by their molecular weight distribution, cationic property, density, and chemical composition [27,28,29,30,31].

In 2015, Tejero et al. prepared polys (methyl methacrylate) using alkynyl alcohols and thiazole azide derivatives as raw materials [27]. The *N*-alkylation of the azole ring allowed the preparation of unipolar and polar polyelectrolytes with different amphiphilic properties, resulting in their antibacterial activity against various microorganisms. Compound **14**, for example, showed significant antibacterial activities, with MIC values less than 10 µg/mL against *P. aeruginosa*, *S. aureus*, *S. epidermidis,* and MRSA (Figure 5).

In the same year, they also prepared a series of compounds with amphiphilic quaternary ammonium salts through controlled quaternization of triazolium side chains by polymethacrylates [28]. The MIC values of compound **15** against *P. aeruginosa* and *S. aureus* were 4 µg/mL at 100% quaternization. Meanwhile, compound **15** with 50% quaternization can also kill 100% of bacteria within 5 min at 2 × MIC. Interestingly, these polymers with a low degree of quaternization still exhibited strong and rapid bactericidal behavior, possibly due to the synergistic effect between the unquaternized heterocyclic and the quaternized heterocyclic (Figure 5).

In 2017, Muñoz-Bonilla et al. prepared a cationic copolymer containing thiazole and triazole groups [29]. They blended it with commercial polystyrene by a simple spin coating method to obtain a series of contact-type active antibacterial films. These blended films exhibited significant microbicidal activity against both G+ and G- bacteria, as well as fungi. At 30% and 50% PS/copolymer content, the killing efficiency of the blended film **16** exceeded 99.99%. In 2018, the researchers reported that even a small amount (3~9 wt %) of PS/copolymer breath figure films prepared using the breath figures approach still had considerable antibacterial effects against G+ bacteria, such as *S. aureus* and *C. parapsilonosis* [30]. The breath figure film **16** was able to kill over 90% of the cells from both bacteria through surface contact (Figure 5).

In 2018, Fernández-García and coworkers synthesized a series of copolymers with quaternary ammonium salt groups that exhibited contact active antibacterial properties [31]. These copolymers were made using quaternary ammonium salts and polyacrylonitrile as raw materials. Compound **17** has extremely strong bactericidal activities against *S. aureus* and *P. aeruginosa*, with cell-killing rates of over 99.99% (Figure 5).

### 3.6. Antibacterial 1,2,3-Triazolium-Based Complex

In 2022, Muñoz-Bonilla and coworkers synthesized compound **18** via free radical polymerization and click reaction with a hydantoin moiety (Figure 6). It underwent further *N*-alkylation and chlorination reactions to form triazolium salt polylactide-based fibers with antibacterial activity. Polylactide-based fiber **18** containing cationic triazolium and *N*-haloamine groups showed good antibacterial activities against both G+ and G- bacteria [32].

To minimize the decreasing effect of ionic bactericidal compounds on the mechanical strength of dental composites, Yeganeh et al. (2017) prepared bactericidal dental nanocomposites containing 1,2,3-triazolium-based functionalized polyhedral oligomeric silsesquioxane (POSS) additives through thiol-one click polymerization. The addition of the triazolium cation resulted in increased bactericidal activity of complex **19** (Figure 6) compared to similar compositions containing dimethyl aminoethyl methacrylate monomers (DMAEMA-BC). The inhibition rate (IR) of complex **19** against *Streptococcus pyogenes* (*S. pyogenes*) was above 60% [33].

Compound **20**, an antimicrobial polyurethane wound dressing film, was prepared through an *N*-alkylation reaction between 1,2,3-triazolium functional soybean oil (TSBO) and methyl iodide (Figure 6). The introduction of 1,2,3-triazolium cation groups into the dressing skeleton increased their antibacterial activity against a range of bacteria. Appropriate concentrations of these cationic groups still maintained the membrane’s good cytocompatibility with dermal fibroblast [34].

### 3.7. Structure-Activity Relationship Analysis of Antibacterial 1,2,3-Triazolium

The cationic properties of triazolium salts are necessary for their antibacterial efficacy. Compounds **1** and **2** have good antibacterial activity, whereas their parent 1,2,3-triazole analogs have no significant antibacterial activity with a MIC value of ≥250 µM [13,14]. The phenolic ketone group in compound **3** is a nonpolar component, which can reduce the hydrophilicity of the triazolium group. There is a balance between the lipophilicity and hydrophilicity of compound **3**, which supports good inhibitory activity against G+ bacteria [15]. However, compound **3** has poor activity against G- bacteria. It is caused by the presence of an outer layer of lipopolysaccharide that hinders the penetration of hydrophobic compounds [15]. Connecting a hydrophobic group at the N-3 and C-4 positions of 1,2,3-triazolium salts is beneficial to the resistance of G+ bacteria. However, excessive increases in hydrophobicity (lipophilicity) will make it more difficult for compounds to pass through bacterial cell membranes, resulting in decreased antibacterial activity (Figure 7).

Therefore, the maximum effectiveness of 1,2,3-triazolium salts against G- bacteria requires the connection of a sufficient but not excessive hydrophobic group. Furthermore, a decrease in hydrophobicity at one substituent position can be compensated by an increase in hydrophobicity at other substituent positions. For example, connecting alkyl chains with different carbon chain lengths at different sites in compounds **5** and **6** maintains their antibacterial effectiveness by maintaining a balance in the overall hydrophobicity of the compounds [17,18].

## 4. Antifungal Activities of 1,2,3-Triazolium Salts

### 4.1. Antifungal 1,2,3-Triazolium-Based Polysaccharide Derivatives

Some studies have revealed that incorporating triazolium groups into starch compounds can enhance their antifungal properties [5,6,7,8,9]. In addition, starch derivatives containing 1,2,3-triazole can be converted to starch derivatives containing 1,2,3-triazolium through the *N*-alkylation process [7]. Starch derivatives containing the 1,2,3-triazolium group exhibited higher inhibitory activities against phytopathogenic fungi than those starch derivatives containing the 1,2,3-triazole group [6,7,8].

In 2017, Guo et al. synthesized a series of compounds with the 1,2,3-triazolium groups. Compound **21** had a good inhibitory effect on the growth of the tested phytopathogens, with an inhibitory index of 82.56% at 1 mg/mL against *Gibberella zeae* (*G. zeae*) (Figure 8). This demonstrated that the introduction of 1,2,3-triazolium groups could improve the antifungal activity of inulin [5].

At the same time, they prepared a series of novel 1,2,3-triazole starch derivatives through copper-catalyzed azide-alkyne cycloaddition (CuAAC). The inhibition rates of these starch derivatives against phytopathogenic fungi (inhibition rates of below 10%) were slightly higher than those of the precursor starches. Cationic starch derivatives with 1,2,3-triazolium side chains (containing different substituents) were also prepared by *N*-alkylation of the 1,2,3-triazole starch derivatives. The antifungal activity of the starch derivative (containing 3-methyl, 4-methanol, 1,2,3-triazolium side chain) **22** showed good activity against phytopathogenic fungi at 1.0 mg/mL (Figure 8). The inhibition rates of **22** against *C. lagenarium*, *F. oxysporum*, and *W. fusarium* were >95%, >70%, and >75%, respectively [6].

Meanwhile, the inhibition rates of compound **23** (starch derivative bearing 1,2,3-triazolium and pyridinium) against *C. lagenarium*, *F. oxysporum*, and *W. fusarium* were >97%, >90%, and >65% at 1.0 mg/mL, respectively (Figure 8) [7].

Additionally, the inhibition rates of compound **24** (starch derivative containing 3- benzyl, 4-ethanol, 1,2,3-triazolium side chain) against *C. lagenarium*, *F. oxysporum* and *W. fusarium* were >90%, >80% and >60%, respectively (Figure 8). The antifungal activities of these 1,2,3-triazolium starch derivatives were superior to those of their precursor 1,2,3-triazole starch derivatives [8].

### 4.2. Antifungal 1,2,3-Triazolium-Based Chitosan Derivative

Chitosan is a promising biocompatible and biodegradable material with broad biological applications as a fungicidal and antibacterial agent. However, its poor solubility in both organic and aqueous solvents severely limit its application. To overcome this limitation, researchers have developed chitosan derivatives bearing the triazolium group, which possess important physical and chemical properties such as water solubility, chemical stability, and antifungal activity [9].

In 2018, Tan et al. synthesized several 1,2,3-triazolium chitosan derivatives and evaluated their biological activities. They found that the antifungal properties of the 1,2,3-triazolium-functionalized chitosan were significantly improved compared to those of 1,2,3-triazole chitosan. Among the tested compounds, compound **25** showed a good inhibition rate of 98.44% at 1.0 mg/mL against *C. lagenarium*, followed by 79.16% and 67.56% against *W. fusarium* and *F. oxysporum*, respectively (Figure 9). Notably, the cationic chitosan derivative **25** remained active against the tested fungi when the concentration of compound **25** was reduced to 0.5 mg/mL. In addition, the chitosan derivatives bearing the 1,2,3-triazolium group showed no cytotoxicity to cucumber seedlings [9].

### 4.3. Antifungal 1,2,3-Triazolium-5-Olates Derivative

In 2018, Glukhareva and colleagues synthesized a series of 1,2,3-triazolium-5-olates, and tested their anti-phytopathogenic fungal activities. Compound **26** showed the best effectiveness against 6 fungal strains (Figure 10), including *Phytophthora infestans* (*P. infestans*), *Cercospora arachidicola* (*C. arachidicola*), *Alternaria solani* (*A. solani*), *G.zeae*, *Sclerotinia sclerotiorum* (*S. sclerotiorum*) and *Rhizoctonia cerealis* (*R. cerealis*), with inhibition rates of more than 80% at a concentration of 50 μg/mL [35].

### 4.4. Structure-Activity Relationship Analysis of Antifungal 1,2,3-Triazolium

Polysaccharide derivatives containing the triazolium group, such as compounds **21** [5,6], **22** [6], **23** [7], and **24** [8], exhibit higher inhibition rates against plant pathogenic fungi than their parent triazole analogs. This indicates that the triazolium group is the key active group. Cationic triazolium may form electrostatic interactions with anionic components on fungal cell walls. In comparison to starch derivatives containing 1,2,3-triazole, the electrostatic interaction of 1,2,3-triazolium derivatives has a greater impact on microorganisms than the hydrogen bond interaction between 1,2,3-triazole analogs and targets. The length of alkyl groups is an important determinant of the antifungal activity of 1,2,3-triazolium functionalized starch derivatives, and their antifungal properties decrease as the side chain length increases. The possible reason for this is that longer alkyl groups tend to provide more electrons to the 1,2,3-triazolium moiety, resulting in a decrease in the positive charge density of the 1,2,3-triazolium moiety and leading to a decrease in antifungal performance. Furthermore, the introduction of pyridine and alkyl groups can enhance antifungal activity (Figure 11).

## 5. Anticancer Activities of 1,2,3-Triazolium Salts

### 5.1. Anticancer 1,3,4-Trisubstituted 1,2,3-Triazolium Salts

By combining triazolium cationic units with fatty chains, a series of compounds with anticancer activities can be obtained [10,36,37]. Many studies have shown that 1,3,4-trisubstituted 1,2,3-triazolium compounds were promising frameworks for developing new drugs for the treatment of cancer patients [10,11,37,39].

In 2016, Osmak et al. synthesized a series of novel anticancer 1-(2-pyridyl)-,4-(2-pyridyl)-,1-(2-pyridyl)- and 4-(2-pyridyl)-3-methyl-1,2,3-triazolium salts, and triazole compounds. The triazolium salt compounds exhibited superior antitumor activities to the triazole compounds in several tumor cells. The cytotoxicity of compound **27** against tumor cells was significantly higher than that of normal cells, and the therapeutic index for lung cancer cells H460 was 7.69 (Figure 12). The mechanism of compound **27** is to block cell mitosis during the G1 phase of the cell cycle. It does not bind to dsDNA but induces reactive oxygen species (ROS) in treated cells, further causing cell death [10].

Two years later, Silva et al. synthesized a series of 1,2,3-triazolium salt compounds with different substituents and tested their inhibitory activities on several cancer cells, including osteomyeloid leukemia HL-60, lymphoid leukemia JURKAT, breast cancer MCF-7 and colon cancer MCT-116. Compound **28** (Figure 12) demonstrated significant anticancer activity compared to other triazolium salts, with a half-maximal inhibitory concentration (IC_50_) value of 3.4 µM against HL-60 cell lines [36].

The IC_50_ values of 1,2,3-triazolium salt **29** against MDAMB-231, 4T1, and HEK-293 cancer cell lines were 60.1, 25.1, and 49.6 µM, respectively, which was superior to Miltefosine (IC_50_ were 191.4, 105.9, and 96.9 µM, respectively). The mixture of compound **29** and Miltefosine had a good anticancer effect against breast cancer cell line HEK-293, with an IC_50_ value lower than 4.2 µM (Figure 12). Compared with the same series of triazolium derivatives, compound **29** had the best water solubility (10 mg/mL), and relatively low toxicity to human peripheral blood mononuclear cells (PBMCs) [37].

In 2019, Antonenko et al. prepared carborane-triazolium cationic salt **30**, which exhibited good anticancer activity against K562 cancer cell lines, with an IC_50_ value of 2.8 µM (Figure 12). Moreover, this boron-containing polyhedral triazolium cationic compound can carry protons through biological membranes, which has potential significance in designing anticancer drugs [11].

### 5.2. Anticancer Anthraquinone Triazolium Compounds

In 2013, Chang et al. demonstrated that the cationic anthraquinone analog compound **31** (4,9-dioxo-1,3-dimethylnaphtho [2,3-d][1,2,3]triazol-3-iu) had good anticancer activities against melanoma, colon cancer, non-small cell lung cancer, and central nervous system (CNS) cancers, with the GI_50_ values of 0.15 to 1.68 µM (Figure 13) [38].

Further, they investigated the anticancer potential of compounds **32** [39] and **33** [20], and found promising activity against A549 cancer cell lines with IC_50_ values of 4.2 and 3.5 µg/mL, respectively (Figure 13). Notably, compound **33** demonstrated remarkable selectivity towards A549 cancer cells relative to human lung normal cells, exhibiting a selectivity index (SI) of 15.09 [20].

### 5.3. Anticancer Allobetulin 1,2,3-Triazolium Derivatives

In 2020, Dehaen and coworkers reported a series of allobetulin derivatives bearing 1,2,3-triazolium, which had better anticancer activities than the parent compound allobetulin and commercial anticancer drug of gefitinib. Compound **34a** showed a good inhibitory effect on SGC-7901 cancer cells, with an IC_50_ value of 1.12 µM. Compound **34b** exhibited broad anticancer activities, especially for HepG2 and Eca-109 cell lines, with IC_50_ values of 1.52 µM and 1.04 µM, respectively (Figure 14) [40].

### 5.4. Anticancer 1,2,3-Triazolium Complex

In 2013, Riela et al. modified the external surface of halloysite with triazolium salts. This resulted in the production of a positively charged halloysite nanotube that was functionalized with triazolium salt **35** (Figure 15). The nanotube is a drug-loading system that has the advantages of high drug encapsulation efficiency and strong controlled, and sustained release capabilities. By utilizing the drug-loading system, the water solubility of two anticancer drugs, curcumin and cardanol, was improved, which in turn overcame their limitations for clinical applications. Furthermore, the drug delivery system could synergize with the two anticancer drugs and enhance their anticancer activities [41,42].

### 5.5. Structure-Activity Relationship Analysis of Anticancer 1,2,3-Triazolium

Aryl substituents can regulate the cytotoxicity of 1,2,3-triazolium salts on tumor cells. Compound **27** with an electron-donating group (4-methoxyphenyl) has better anticancer activity compared to the compounds bearing electron-neutral groups (e.g., phenyl) or electron-withdrawing groups (e.g., 4-(trifluoromethyl)phenyl) [10]. Compounds **28** and **29** bearing decyl substituents at the N-1 positions exhibit strong cytotoxicity against cancer cells. Interestingly, there is no significant difference in the effect of different anionic forms of the same triazolium salts on anticancer activities [36,37]. In addition, the 1,2,3-triazolium group is an effective mitochondrial targeting group [11,37]. Research studies suggest that 1,2,3-triazolium salts induce cancer cell apoptosis through mitochondrial apoptosis and cell cycle arrest pathways [11]. This may be due to the stronger mitochondrial electronic effect in cancer cells than in normal cells [37]. Furthermore, in complex living cell systems, the structure–activity relationship of 1,2,3-triazolium salt compounds is the result of a combination of stereoelectronic effects (Figure 16) [37].

## 6. Antileishmanial Activities of 1,2,3-Triazolium Salts

### 6.1. Antileishmanial 1,3,4-Trisubstituted 1,2,3-Triazolium Salts

In 2017, Silva et al. synthesized a series of 1,3,4-trisubstituted 1,2,3-triazolium salts, and tested their effects against *Leishmania amazonensis* (*L. amazonensis*). Compound **36**, which contained an acetate anion and linked with a side chain of 10 carbon atoms at the N-1 position and a methyl group at the N-3 position, exhibited highly selective biological activity against *L. amazonensis* (Figure 17). Its IC_50_ value was close to 1.0 µM in the intracellular amastigotes (stages of the parasite related to human disease), which was lower than the control drug Miltefosine (IC_50_ = 4.2–22 µM). In addition, compound **36** showed non-cytotoxic effects on human red blood cells and macrophages (half-maximal cytotoxic concentration, CC_50_ = 115.9 µM), and was more destructive to intracellular parasites (SI > 115) [43].

Three years later, they synthesized a series of 1,2,3-triazolium salts, and tested their inhibitory activities against *L. amazonensis* (the forms of promastigotes and intracellular amastigotes). Compound **37** (Figure 17) exhibited good activities against the promastigotes (IC_50_ = 3.61 µM) and intracellular amastigotes (IC_50_ = 7.61 µM) of *L. amazonensis*, which was superior to the control drug Miltefosine (IC_50_ = 15.05 and 11.5 µM, respectively) [44]. Additionally, compound 37 has good selectivity between intracellular amastigotes of *L. amazonensis* and Macrophages with a SI of 10.07.

In 2022, Coimbra et al. prepared seven novel 1,2,3-triazolium salts and investigated their inhibitory activities against *Leishmania infantum* (*L. infantum*) OP46, MCAN1112, and ITMAP-263 strains. The IC_50_ values of compound **38** for promastigotes of the 3 strains were 5.28, 5.57, and 4.55 µM, respectively, which were lower than or similar to the control drug Miltefosine (IC_50_ values of 2.80, 6.48, and 7.81 µM, respectively). In addition, compound **38** exhibited a low cytotoxic effect on macrophages, with a CC_50_ value of 47.92 µM (Figure 17). In the inhibition test against intracellular amastigotes, compound **38** also showed good activity (IC_50_ = 7.92 µM), with an SI of 6.05 [12].

### 6.2. Antileishmanial Other 1,2,3-Triazolium Salts

In 2022, Velázquez and colleagues reported that 1,2,3-triazolium salts exhibited higher antileishmanial activity in intracellular amastigotes than their parent triazoles. Compound **39**, which had a biphenylethyl substituent on the triazolium cation group, exhibited an EC_50_ (half maximal effective concentration) value of 4.8 μM against *L. infantum* axenic amastigotes (Figure 18). It also showed good selectivity between Leishmania and macrophages, with an SI above 10.3. In addition, compound **39** significantly decreased the content of low molecular weight mercaptans in intracellular amastigotes of *L. infantum*, and LiTryR may be the main target of this new compound [45].

In 2017, Sanchez-Moreno et al. synthesized several new [1,2,3]triazolo[1,5-a]pyridinium salts using triazolopyridine compounds as raw materials, and studied their in vitro antileishmanial activities. The results showed that compound **40** had good activities against intracellular amastigotes and promastigotes of 3 species of Leishmania, including *L. infantum*, *Leishmania braziliensis* and *Leishmania donovani*, with IC_50_ values of 5.3–13.4 μM and 6.3–9.1 μM, respectively (Figure 18). The activities were better than that of the control drug Glucantime, with IC_50_ values of 18.0–25.6 μM and 18.4–30.4 μM, respectively. The selectivity index (SI > 56) of compound **40** was higher than the control drug Glucantime (SI < 1) [46].

### 6.3. Structure-Activity Relationship Analysis of Antileishmanial 1,2,3-Triazolium

Compounds **36**, **37**, and **38** bearing 10-carbon side chains at the N-1 position and a methyl group at the N-3 position exhibit good antileishmanial activities [12,43,44]. The long carbon chain at the N-1 position of these compounds is beneficial to their antileishmanial activities. The analogs of compound **38** containing 12, 14, or 16 carbon atoms in the alkyl side chain exhibit better antileishmanial activities, but the long carbon chain also leads to strong toxicity against macrophages [12]. Furthermore, those 1,2,3-triazolium salts bearing a propyl group on the N-3 position exhibit a strong toxic effect on macrophages, regardless of the length of the side chain [12]. It is worth noting that the 1,2,3-triazolium group is crucial for enhancing the antileishmanial activities of the compounds, since compounds **37** and **38** exhibit higher antileishmanial activities than their parent triazole [12,44]. Different anion types of the compounds have certain effects on antileishmanial activities and selectivity. For example, compound **36** contained acetic acid anions and was 5.5 times more active against promastigotes of *L. amazonensis* than analogs containing iodine anions and exhibits highly selective biological activity (Figure 19) [43].

## 7. Conclusions

In summary, 1,2,3-triazolium salts can be directly synthesized in three steps, including CuAAC, *N*-alkylation, and salt metathesis. A broad scope of 1,2,3-triazolium derivatives bearing various substituents and substitution patterns can be achieved through these approaches. In addition to their impressive applications as ionic liquids (ILs), catalysts, and metal ligands. 1,2,3-triazolium salts exhibit a broad range of biological activities, such as antibacterial, antifungal, anticancer and antileishmanial properties. They are valuable in the development of pesticides and pharmaceuticals. Incorporating 1,2,3-triazolium groups into polymer materials via simple synthetic methods can generate antibacterial properties. The utilization of such materials can address the issue of microbial contamination in medical devices or specialized antibacterial environments. Additionally, 1,2,3-triazolium can be incorporated with nanomaterials, such as compounds **19** and **35**, imparting strong hydrophilicity to enhance the ductility of nanomaterials and antibacterial properties. The incorporation of 1,2,3-triazolium salt compounds into anticancer drug carriers can bring additional possibilities. Modifying halloysite nanotubes with triazolium salts can yield positively charged anticancer drug carriers possessing improved encapsulation efficiency, controllable release ability, increased water solubility, and thus enhanced anticancer activity. These polymer derivatives can not only be employed to generate antibacterial films through polymerization, but can also be used as additives or components in fibers and nanomaterials.

Based on the structure–activity relationship of 1,2,3-triazolium salt derivatives, it is believed that substituents at N-1 and N-3 positions play critical roles in biological activities. The lip/water balance of 1,2,3-triazolium salts is critical to their high biological activity and selectivity. Additionally, 1,2,3-triazolium derivatives may have multiple action modes, and electrostatic interactions with anionic components on the cell membrane are among the most important. Compared with the parent triazole, 1,2,3-triazolium salts have superior biological activities and water solubility, and are promising in the development of new drugs for drug-resistant pathogens. Currently, the biological activities of 1,2,3-triazolium salts are receiving increasing attention from researchers. More precise and reliable data are required to elucidate the possible action mechanism of triazolium salts, providing a theoretical basis for their design and synthesis. Overall, 1,2,3-triazolium salts have promising development prospects in the fields of agriculture, medicine, and industrial production.

## Data Availability

Not applicable.

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
