# Peer review of "Research Advances on the Bioactivity of 1,2,3-Triazolium Salts"

_ijms, 2023, doi:10.3390/ijms241310694_

Round 1

Reviewer 1 Report

In the present manuscript the authors provide a critical review on the synthetic advances and bioactivity studies of 1,2,3-triazolium salts. The manuscript flows well and is clearly organized into sections. The molecular structures and the schemes are clear and highlighted using different colors. The manuscript provides a clear picture of the last achievements on the application of 1,2,3-triazolium salts as antibacterial, antifungal and anticancer agent in different matrixes concisely and clearly. Finally, the quality and number of references is appropriate for a short review. Nevertheless, there are few points that require to be addressed before recommending publication.

In section 2.2 many words are connected by dashes, please fix it

Page 3 line 108 “Quaternary ammonium salt compounds (QACs) have a cationic head and a long lipophilic tail that is similar to phospholipids found in cell membranes [45].”  This sentence is not completely correct. tetramethyl ammonium chloride is a QAC but does not have any long lipophilic tail.  Some QACs might have long lipophilic tails mimicking phospholipids structures but not all of them are. Please revise the definition and the statement accurately.

Page 4 line 139 “Anthraquinone has the structural core of anthracycline” should be revised as “Anthraquinone is the structural core of anthracycline” since anthraquinone is a specific molecular scaffold of many natural compounds including anthracycline.

Page 5 line 173 “Amphiphilic cationic peptides, also known as N-substituted glycine oligomers, show great prospects as antimicrobial peptide mimetics.” This sentence is not completely correct. N-substituted glycine oligomers, also known as peptoids are a specific family of amphiphilic cationic peptides, which are many and different and not always constituted by glycine derivatives. Please revise the definition and the statement accurately.

Page 14 line 484 “Compounds 36, 37, and 38 bearing a 10C side chain” 10C should be revised as 10 carbons or decamer.

Author Response

Response to Reviewer 1 Comments

Point 1: In section 2.2 many words are connected by dashes, please fix it.

Response 1: We have corrected these issues on Page 2, lines 65, 69, 74 and Page 3 line 86 of the revised manuscript.

Point 2: Page 3 line 108 “Quaternary ammonium salt compounds (QACs) have a cationic head and a long lipophilic tail that is similar to phospholipids found in cell membranes [45].” This sentence is not completely correct. tetramethyl ammonium chloride is a QAC but does not have any long lipophilic tail. Some QACs might have long lipophilic tails mimicking phospholipids structures but not all of them are. Please revise the definition and the statement accurately.

Response 2: We thank this referee for his valuable suggestions. We have checked and corrected the definition of the “Quaternary ammonium salt compounds (QACs)” on Page 3 line 109 in the revised manuscript.

Point 3: Page 4 line 139 “Anthraquinone has the structural core of anthracycline” should be revised as “Anthraquinone is the structural core of anthracycline” since anthraquinone is a specific molecular scaffold of many natural compounds including anthracycline.

Response 3: We have changed “Anthraquinone has the structural core of anthracycline” to “Anthraquinone is the structural core of anthracycline” in the revised manuscript.

Point 4: Page 5 line 173 “Amphiphilic cationic peptides, also known as N-substituted glycine oligomers, show great prospects as antimicrobial peptide mimetics.” This sentence is not completely correct. N-substituted glycine oligomers, also known as peptoids are a specific family of amphiphilic cationic peptides, which are many and different and not always constituted by glycine derivatives. Please revise the definition and the statement accurately.

Response 4: We have accurately revised the definition of the “Amphiphilic cationic peptides” on Page 5 line 173 of the revised manuscript.

Point 5: Page 14 line 484 “Compounds 36, 37, and 38 bearing a 10C side chain” 10C should be revised as 10 carbons or decamer.

Response 5: we have modified the “10C” to “10 carbons” on Page 15, line 495 of the revised manuscript.

Author Response

Response to Reviewer 2 Comments

Point 1: Please ensure that the full form of the abbreviation is given at the first occurrence.

Response 1: We have carefully checked and corrected the full form of the abbreviation in the revised manuscript.

Point 2: A figure with SAR representation for the chemical structures at the end of each chapter has to be included, highlighting, if possible, the role of the triazole moiety in the recognition of the target.

Response 2: We have added the figure with SAR representation for the chemical structures at the end of each chapter. This information has been provided in the revised manuscript (Page 8 line 277-278, Page 11 lines 361-362, Page 14 lines 443-444 and Page 15 lines 509-510).

Point 3: Screening of the manuscript for typographical errors is required. For instance, In the 3.7-chapter, line number 269, the phrase “against G- bacteria. it is caused” need to be checked and changed as “against G- bacteria. It is caused”.

Response 3: We have corrected this error in the revised manuscript.

Point 4: The structure of compound 24 (Figure 7, Page 9) is not in accordance with the name reported in the text, Page 9, lines 308-309.

Response 4: We have corrected the structure name of the compound 24 in the revised manuscript, Page 9 lines 314-315.

Point 5: The IC50 of compound 34a reported in the Figure 12, Page 12, does not match with that reported in main text, Page 12, line 408. The same for compounds 36 (Figure 14, Page 13) and 39 (Figure 15, Page 14).

Response 5: We have corected the IC50 of compound 34a in the revised manuscript. Furthermore, we have added the IC50 of compound 36 against intracellular amastigotes in the Figure 17, Page 12 line 472. And we corrected the IC50 of compound 39 in the revised manuscript, Page 14 line 478.

Point 6: The format of the references needs to be checked.

Response 6: We have checked and corrected the format of the references.

Point 7: Check reference 30; the year of the paper it is not in accordance with that reported in the text, chapter 3.5, Page 7, line 232.

Response 7: We have checked and corrected the year in the revised manuscript, Page 7, line 230.
